# Efficient Detection and Tracking of Human Using 3D LiDAR Sensor

**DOI:** 10.3390/s23104720

**Published:** 2023-05-12

**Authors:** Juan Gómez, Olivier Aycard, Junaid Baber

**Affiliations:** Laboratoire d’Informatique (LIG), University of Grenoble Alpes, 38000 Grenoble, France; djcampo@uninorte.edu.co (J.G.); olivier.aycard@univ-grenoble-alpes.fr (O.A.)

**Keywords:** 3D point cloud, person detection, tracking, classification, real-time

## Abstract

Light Detection and Ranging (LiDAR) technology is now becoming the main tool in many applications such as autonomous driving and human–robot collaboration. Point-cloud-based 3D object detection is becoming popular and widely accepted in the industry and everyday life due to its effectiveness for cameras in challenging environments. In this paper, we present a modular approach to detect, track and classify persons using a 3D LiDAR sensor. It combines multiple principles: a robust implementation for object segmentation, a classifier with local geometric descriptors, and a tracking solution. Moreover, we achieve a real-time solution in a low-performance machine by reducing the number of points to be processed by obtaining and predicting regions of interest via movement detection and motion prediction without any previous knowledge of the environment. Furthermore, our prototype is able to successfully detect and track persons consistently even in challenging cases due to limitations on the sensor field of view or extreme pose changes such as crouching, jumping, and stretching. Lastly, the proposed solution is tested and evaluated in multiple real 3D LiDAR sensor recordings taken in an indoor environment. The results show great potential, with particularly high confidence in positive classifications of the human body as compared to state-of-the-art approaches.

## 1. Introduction

Recent advancements in LiDAR technology have revolutionized many industries, making it more reliable and accurate for various applications such as autonomous driving, human–robot interaction, and more [1,2]. Three-dimensional LiDAR data provide a comprehensive representation of the environment in 360 degrees, which can be used by robots or autonomous cars to make informed decisions. The applications of 3D LiDAR are extensive, including autonomous driving [2], forestry [1], medical training [3], smart city deployments [4], remote sensing [5], and 3D SLAM [6,7,8].

However, it is worth noting that 3D LiDAR sensors are costly, making it challenging to mount them on multiple mobile robots or systems. Even the CEO of Tesla has reservations about using 3D LiDAR for autonomous driving due to its cost (https://www.youtube.com/watch?v=BFdWsJs6z4c, accessed on 8 May 2023). As a result, static sensors, which are mounted in a fixed location, have emerged as a potential solution to this problem.

Furthermore, static sensors can be used in conjunction with mobile robots or systems, making them more efficient and effective. By mounting 3D LiDAR in a suitable location, different mobile robots can access the data and make informed decisions. This approach can significantly reduce the cost of deploying multiple 3D LiDAR sensors on each mobile robot or system, making it more feasible and affordable.

In summary, while the applications of 3D LiDAR sensors on mobile robots are extensive (Figure 1), static sensors have emerged as a potential solution due to their cost-effectiveness and versatility. By highlighting the potential applications of static sensors, we can encourage further research and development in this area, leading to more innovative solutions for various industries.

In the field of robot perception, the Detection and Tracking of Moving Objects (DATMO) is one of the main problems in almost every robotic application, such as autonomous driving or human-robot interaction. A great deal of previous work has been done to tackle this particular problem in perception, most of which has used depth or 3D sensors such as RGBD and stereo cameras.

The RGB-D cameras provide both RGB information and depth data. RGB information could be useful in general vision applications such as detection by skin detectors and also for specific applications such as person re-identification. Some examples of the possible applications can be the segmentation and classification of the legs [9] and histograms for the height difference and colors [10]. Moreover, multiple detectors could be implemented depending on the object’s distance [11]. In addition, range data such as a target height can be used in order to improve the accuracy [12]. The main limitations of RGB-D cameras are the range of field view, which is limited, and RGB-D and stereo cameras are slow to extract depth information and the results are often not precise [13,14]. Two-dimensional LiDAR is used for depth information for various applications such as autonomous driving and human tracking by the robot [15,16,17]. The 2D LiDAR sensors are affordable and precise, and 2D point clouds are not computationally expensive to process. The main limitation of 2D LiDAR is that it perceives only one plane of the environment which makes it difficult to detect people with high confidence. Meanwhile, 3D LiDAR offers a high resolution of points at a high speed with great precision, resulting in a very descriptive geometry of the environment [14,15,18,19]. Objects in 3D LiDAR point-clouds can be described geometrically as they appear in the real world, with accurate representations of length, width, and height up to a certain precision. The 3D LiDAR generates a huge amount of data per scan. In order to handle the high amount of data obtained with 3D LiDARs and achieve a real-time implementation, prior 3D maps have been used for background subtraction to reduce the computation time [16], but this requires previous knowledge of the environment. In contrast, in our approach, we rely on a background subtraction algorithm without any previous knowledge of the environment, and our solution does not rely on a first movement to detect a person. Overall, for person detection (even at close range), the data provided by a 2D LiDAR sensor is far too limiting to accomplish the ultimate task of analyzing and understanding a person’s behavior.

For DATMO problems, occlusion is one of the major problems with all sensors, particularly when the object/robot is stationary. There are attempts to address occlusion such as tracing the rays from positions near the target back to the sensor [20].

Previous works on person detection and tracking often rely on movement detection and clustering, which can be challenging. Our proposed framework simplifies this process by focusing solely on person detection and tracking using 3D LiDAR. In our solution, classifying moving objects is a crucial step because our model places high priority on them. Deep learning supervised models have been used for person detection by classifying point clouds directly, which could replace the movement detection module [21]. However, the process of classifying the whole frame can be computationally expensive.

Object detection and tracking using R-CNNs have also been explored, but these studies have been limited to cars only [22,23].

It should be noted that in addition to the proposed framework, there are other existing frameworks that use 3D LiDAR technology, such as VoxelNet [24], OpenPCDet [25], RCNN [26], HVNET [27], and 3D-Man [28], which are primarily focused on outdoor objects such as cars, pedestrians, and cyclists. However, these deep learning models require the use of GPUs and are not yet capable of real-time processing. While the PV-RCNN++ framework, also known as OpenPCDet, is faster than its predecessors, it still cannot be used on CPU-based machines. Moreover, detecting a person in an indoor environment is far more challenging than detecting a pedestrian on the road, as the former may have more complex and varied positions that must be accurately detected and tracked.

A 3D LiDAR sensor such as Ouster generates up to 64 K points per scan on average. Processing all these points in real-time applications is discouraged, so only a subset of points are extracted and processed [23].

Our work presents an efficient solution for person detection, classification, and tracking. Our proposed framework consists of several modules, namely movement detection, voxelization, segmentation, classification, and tracking.

Our solution has several notable features, including a modular structure comprising multiple stages that work together synergistically to achieve optimal performance and results. Moreover, there is a continuous interaction between classification and tracking, which allows for a robust and consistent performance when dealing with extreme pose changes such as crouching, jumping, and stretching, or even with some of the sensor’s limitations.

Furthermore, our solution can be implemented in real-time on low-performance machines, making it highly versatile and practical for detecting, classifying, and tracking individuals in indoor environments. Overall, our work provides a highly effective solution for the accurate and reliable detection, classification, and tracking of persons, which has a wide range of applications in various fields.

## 2. Implementation and Challenges

We used the Ouster OS1-32 3D LiDAR sensor for our solution, which has a vertical resolution of 32 layers and customizable horizontal field of view up to a full 360-degree turn. The base frequency of operation is 10 Hz, and the minimum and maximum ranges of the sensor are 0.8 m and 150 m, respectively. The sensor’s precision is 1.1 cm in the 1–20 m range, and it is shown in Figure 2a.

Although the maximum sensor range is 150 m, it becomes challenging to differentiate a moving object from the background at greater distances due to the sensor’s resolution. To take advantage of the descriptive geometry that the sensor provides, we need to have the maximum number of points possible on our objects of interest. Therefore, our solution is designed for small indoor ranges of up to 8 m in any direction. We evaluated our solution in two different offices: one big office of up to 12 m in length and 5 m in width, and a small office space of 4 m in length and 5 m in width. In both cases, the sensor was positioned at a height of 1.2 m, simulating a top-mounted sensor on a mobile robot platform.

### 2.1. Restricted Vertical Field of View

Despite having a complete 360-degree horizontal field of view, the Ouster OS1-32 sensor is restricted to only 33.2 degrees of vertical field of view. This restriction can cause objects to appear chopped in the resulting scan, especially when they are close to the sensor. Since our solution is designed for small indoor environments, people may walk close enough to the sensor to appear chopped in the final scan. This phenomenon presents a challenge to the classifier since it relies on object dimensions and shape. Figure 2b,c shows an example of this situation.

### 2.2. Extreme Pose Change and Occlusion

Extreme pose change is a challenge for any person detector and tracker as people can assume a wide variety of poses, making it difficult for a classifier to detect them in all possible shapes. Poses such as crouching, stretching, jumping, and picking up objects are some of the examples. To address this issue, our solution uses an interaction between the classification and tracking modules to overcome their shortcomings.

Another common problem encountered in tracking is target occlusion [20]. Having a sensor at a specific position generates the problem of moving objects being occluded by static objects or each other. Although our solution does not deal explicitly with occlusions, we evaluated it to see how it behaves in their presence.

## 3. Architecture and Implementation

Figure 3 shows the flow chart of the proposed solution where each module is differently numbered. The first module receives a 3D point-cloud from the sensor and performs movement detection to find ROIs. The ROIs return a significantly smaller number of points to the process, which saves computation time and makes 3D point-cloud processing in real-time possible. The ROIs are then passed for voxelization and segmentation, respectively. These segments, also known as objects, are then passed to the classifier. Finally, the last module, the tracking module, handles the creation, updating, and elimination of the tracks.

### 3.1. Movement Detection

As already mentioned, in order to achieve a real-time implementation, we need to process the smallest number of points possible at each scan. In an indoor environment, i.e., an office, most of the environment is static, which is not the case in outdoor environments. Rather than processing all the points, which will be redundant, only ROI regions are processed, which are detected by movement detection. For a sensor at a static position, the most common and efficient way to extract the ROIs is the process of background subtraction. Therefore, the first step is the creation and calibration of the background. This process is executed at the start of the algorithm; we process all points in the scan to create the background, as illustrated in Figure 3. In this way, if there is a person in the environment at the moment of the creation, we eliminate those points from the final model. After the background model creation, we assume all the points in a scan as static points, and ROIS are extracted by comparing the subsequent scans.

### 3.2. Voxelization and Segmentation

The 3D point cloud obtained from the sensor consists of *x*-, *y*-, and *z*-coordinates in the Cartesian space. Unlike RGB images, the raw point clouds are unstructured and contain no semantic information about the scene. Therefore, all these data cloud points are represented by a higher level, known as a voxel [29]. Voxelization is a widely practiced technique in 3D point-cloud processing and the structuring of cloud points. In the proposed framework, we have used the implementation of Trassoudaine et al. [23] for voxelization and segmentation.

### 3.3. Classification

After the segmentation process, we have the list of segmented objects (formed by voxels) that are inside the ROIs. The goal of the classifier is to solve a binary classification problem, i.e., either the given object is a person or the background. To improve the true positives in the proposed framework, we only focus on moving objects, as shown in Figure 4. In the proposed framework, three weak classifiers are taken, and each weak classifier has its own criteria for the classification. In the end, each vote for the objects. Since, there are three classifiers and two classes, therefore, based on a majority vote, the decision is made for the final label for the given object. The weak classifiers are discussed later in this section.

#### 3.3.1. Shape Classifier

As already mentioned, one of the main benefits of 3D LiDAR with high resolution is that we can extract precise and multiple geometric features from the objects. The shapes classifier takes advantage of the descriptive geometry provided by the sensor. It is a simple classifier that compares the dimensions and shape of each object to one of the “strict” geometric models of a person that we already have. These models represent a variety of attributes that an object should have to be considered as a person. These attributes include width, height, length, and the proportions between them. For example, if an object is taller than 2.3 m, it can easily be concluded that it cannot be a person, or if an object has almost the same width and height, it cannot be a person. With a variety of these simple thresholds/criteria, this classifier can recognize a person when they are in their most basic poses (standing up or walking). The idea is to only start the tracking when we have high confidence that an object is a person. Therefore, our classifier cannot deal directly with the majority of poses a person can undergo other than walking and standing up; the integration between the classification and tracking can overcome this problem, as discussed in the tracking module.

#### 3.3.2. Normals Classifier

Even though the shape classifier is made to recognize a person under the most basic circumstances, its simplicity might also provide false negative or false positive results. In order to fix this, we use the normal vector classifier. For every object, we calculate the normal vectors of each voxel that form them. Estimating the surface normals in a point cloud is a problem of analyzing the eigenvectors and eigenvalues or PCA (Principal Component Analysis) [23], and this can easily be done in an ROS (robot operating system) using PCL (Point Cloud Library). Once we have the normals, we calculate the average contribution of each of the components of the vectors. Therefore, if the object has most of its normals parallel to the ground, it is most likely a wall or an office division. In this way, we can eliminate most of the false positives that result from a portion of a wall or division that has dimensions similar to that of a person.

#### 3.3.3. Shade Classifier

Lastly, this classifier was made to correct possible misclassifications that result from an object of the background being partially occluded by a moving object. When this happens, the shape of the object could be altered enough to be considered a person, and the normal vectors calculation could also be affected, because of the smaller number of points. This is what we called the shade effect. In order to avoid this, we created a weak classifier that checks if an object is being affected by this effect by tracing the rays close to the object back to the sensor—if the rays hit another object in the way, then there is a shadow. The situation is explained graphically in Figure 5. This process is similar to the one presented in [20], where it was done to detect partial and full occlusions of the tracked object.

Finally, all three classifiers vote for the given object. Based on the majority voting, the object is classified as human or the background.

### 3.4. Tracking

Figure 6 shows the flow diagram for the tracking module. This module is responsible for creating, eliminating, and updating the tracks. Moreover, it also handles the motion prediction stage where a predicted ROI is created for each tracked person. Finally, it can also filter some of the possible mistakes in the classification module.

#### 3.4.1. Track Creation and Elimination

The classification module gives good accuracy; however, in some cases, there are some false positives. To handle the false positives in tracking, we ensure that a track is only created when the same object is classified as a person by the classification module in three consecutive scans, and a track is eliminated if the tracked person is not found after 20 consecutive scans (at 10 Hz, this would be 2 s).

#### 3.4.2. Motion Prediction

The motion prediction method also creates ROIs by keeping track of the previous object’s positions based on the velocity. Assuming a constant velocity, we predict the next position of the target, and we create a gating area of 0.5 m around the predicted position (in every possible direction).

The radius of the gating area corresponds to the maximum distance a person can travel at a maximum speed in the time before the next scan. The ROI becomes one of the inputs to the next time step from the voxelization module, as shown in Figure 3.

#### 3.4.3. Track Updation

At every time step, every ROI goes through the voxelization, segmentation, and classification process. The result is a list of classified objects in the ROIs only. The objects that are classified as persons are assigned to their corresponding tracks—this is similar to GNN (global nearest neighbor) for data association. Basically, the person closest to the last recorder position of the track is assigned to it. Later, the framework tries to update the classification by removing the false positives and false negatives, as shown in Figure 7.

## 4. Validation and Results

To evaluate the system, *precision*, *recall*, and *F*-measure are used. These are widely used metrics in robotics literature [13,30,31]. The videos are recorded while positioning the sensor at the height of 1.2 m, which is set to mimic a top-mounted sensor on a mobile robot platform. To show the strength of the framework, several cases are created, as illustrated in Table 1.

For cases from 1 to 4, the sensor is placed in an office with a length of 10 m and a width of 6 m. In this experiment, cases are categorized from basic to challenging, gradually. In the case 1 recording, a single person is walking around the room normally; in case 2, a single person is walking where a big obstacle was placed in the middle of the office to simulate a large amount of occlusion; in case 3, a single person is doing extreme pose changes such as crouching, stretching, jumping, and doing pick and place actions; and in case 4, two persons are walking around in the office. The results can be seen in Table 2 (videos of the results can be found at the URL https://lig-membres.imag.fr/aycard/html//Projects/JuanGomez/JuanGomez.html, accessed on 8 May 2023).

As expected, case 1 achieves the highest *F*-score, since it is the simplest case. The person is correctly tracked at all times, with only a few false negatives. Some false positives can also be seen due to the shadows created when the person is close to the sensor.

The false positives have an impact on the metrics; however, they marginally affect the tracking. This is because of the constraint that is imposed to start the tracker after three consecutive classifications, as false positives or negatives mostly do not last in consecutive scans.

As seen in Table 2, in the case of complex *poses*, the proposed framework is indeed able to keep the tracking consistent, as shown in Figure 8. This is possible due to the interaction between the classification and tracking modules, as explained in Figure 7. Even though the occlusion is not explicitly handled in the implementation, in the results, it can be seen that the recall is affected by the number of false negatives (due to the occlusions), whereas the precision remains stable. Interestingly, the proposed framework remains effective even for more than one person. Despite the fact that our solution was mostly intended for single-person tracking, it shows potential for multiple-person tracking as well. The metric that decreases the most is the computing performance, outputting at 6.8 Hz down from the 10 Hz of the input frequency. This is normal since the more persons there are in the scan, the more points the algorithm has to process.

In experiments with more challenging cases, such as case 5, the system remains effective. For case 5, the sensor is placed in a smaller part of the building, which corresponds to a small rest area surrounded by halls. Overall, we achieve good results in every situation. The cases included the persons walking around, sitting in a chair, and occasionally crouching to tie their shoes. The poses included three that were either moving, jumping, or doing aerobics moves. The qualitative results can be seen online where multiple processed videos are shown. The repository (https://github.com/baberjunaid/3D-Laser-Tracking, 8 May 2023) contains several bag files, scripts, and ROS packages online for future experiments and re-implementation. Table 3 provides a comparison between our proposed implementation and other recent LiDAR-based scene segmentation frameworks. To reduce the computational burden of processing the entire 3D scan, we extract the Region of Interest (ROI) using an adaptive clustering approach [13]. This approach achieves high recall by detecting various objects in the room, such as chairs, boxes, and walls. However, the detection of numerous objects results in a decreased precision of the framework. On average, our proposed framework takes 0.55 s to process a scan on a core i5 with 8 GB RAM.

The online learning framework [31] also utilizes an adaptive clustering framework for ROI extraction and later extracts seven features to train the classifier. As a result, our customized configuration yields the same recall as the online learning framework, as presented in Table 3.

Table 4 shows the bandwidth usage of a ROS node, measured using the **rostopic bw** command. The table reports the average bandwidth usage, as well as the mean, minimum, and maximum bandwidth usage values for a window of 100 measurements. In the case of the baseline method, the average bandwidth of the topic is 16.59 MB/s, which means that on average, data are being sent at a rate of 16.59 megabytes per second. The mean bandwidth over the period of measurement is 1.69 MB/s. The minimum bandwidth measured over the period of measurement is 1.67 MB/s, and the maximum bandwidth measured over the period of measurement is 1.72 MB/s. The window value of 100 indicates that the average, mean, min, and max values were computed over the last 100 measurements. In the context of the output from **rostopic bw**, the *average* refers to the overall average bandwidth usage over the entire measurement period, while the *mean* refers to the average bandwidth usage per message.

The average bandwidth usage can be large compared to the mean if there are periods of time during the measurement where the bandwidth usage is higher than the overall mean. For example, if there are short bursts of high bandwidth usage interspersed with longer periods of low bandwidth usage, the average bandwidth usage can be skewed upwards by the bursts.

It is also worth noting that the *mean* value reported by **rostopic bw** may not be a very reliable metric, since it is calculated based on the number of messages and the total bandwidth used over the measurement period, and these values can fluctuate rapidly. The *average* value is likely to be a more stable and reliable metric for assessing the overall bandwidth usage.

The proposed framework has a main limitation in that it is designed specifically for detecting and tracking humans in a closed/indoor environment where the LiDAR sensor is stationary. It may not perform as well in situations where the sensor is mounted on a mobile robot or when there are many objects in the outdoor environment, as the increased range of possible objects and shapes can confuse the classifier. In such cases, a combination of adaptive clustering and the proposed framework may be necessary, but this can be more time-consuming. Therefore, the proposed framework is best suited for indoor settings where the environment is relatively stable and the focus is on detecting and tracking human movements.

## 5. Conclusion and Future Works

In this paper, we have developed a framework for detecting and tracking individuals in 3D point clouds in near real-time on standard hardware. Our approach involves a robust object segmentation technique using super-voxels and a chaining method, followed by an accurate classification of the objects, integrated with a tracking mechanism that effectively handles extreme pose changes of individuals. Our experimental results demonstrate the effectiveness of our prototype in detecting and classifying individuals even in challenging scenarios and occlusions, as shown in Figure 8. The further development of our approach on a larger scale holds potential for various applications in the field of social robotics and human-robot collaboration, especially with the inclusion of geometric methods to detect and segment individuals into multiple objects such as arms, heads, etc., for applications such as industrial human–robot collaboration and smart surveillance. The framework is developed in an ROS environment and can be easily used with any distribution of ROS.

## Figures and Tables

**Figure 1 sensors-23-04720-f001:**
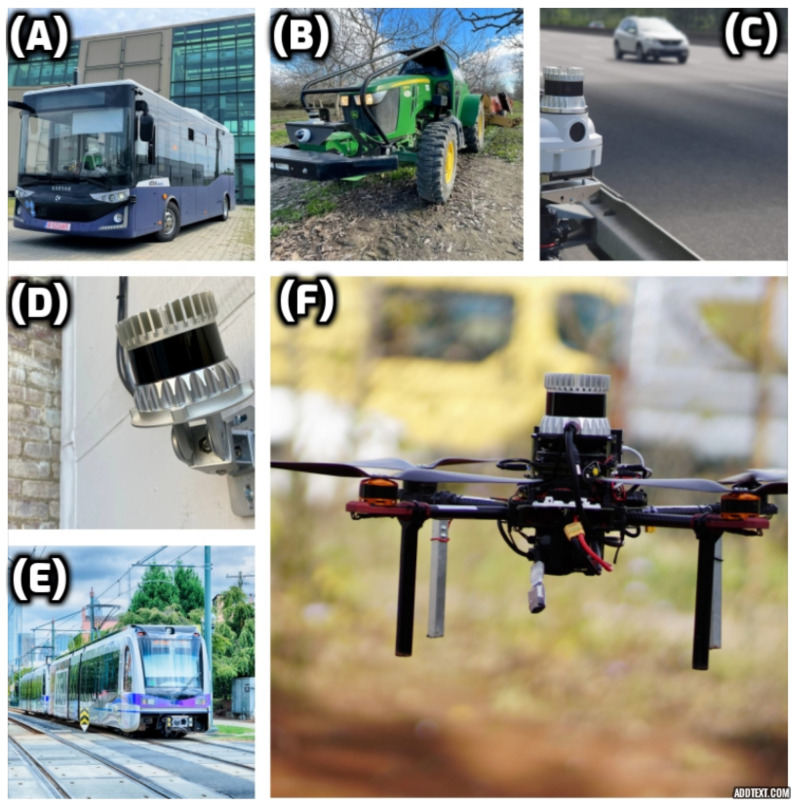
Applications of 3D LiDAR in industry, smart infrastructure, and automotive areas. (**A**) The smart system for bus safety operating on public streets, (**B**) applications in agriculture, (**C**) intelligent transportation, (**D**) intelligent security management, (**E**) intelligent rail monitoring and inspection, and (**F**) drones and mobile robots.

**Figure 2 sensors-23-04720-f002:**
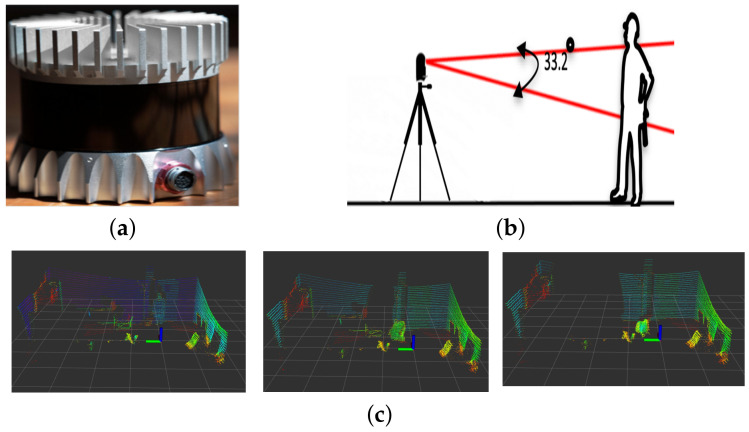
Some limitations of a 3D LiDAR sensor in indoor environments: (**a**) Ouster sensor used in the experiment, (**b**) example of the restricted vertical field of view leading to chopped scans, and (**c**) severe occlusion of a moving object due to the sensor’s fixed position, the different shades of the color show the reflectance values of the returned laser signals.

**Figure 3 sensors-23-04720-f003:**
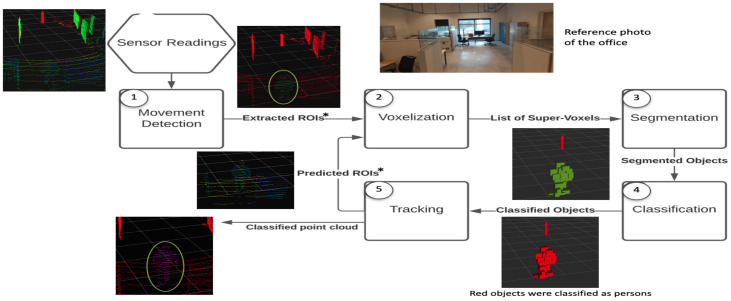
Flow diagram of the proposed system showing the connections between each module. The sensor position is represented by the red arrow, static points are represented by red points, dynamic points by green points, and points classified as part of a person by pink points (as highlighted in the green circle). Regions of interest (ROIs) are generated by movement detection denoted by the * symbol.

**Figure 4 sensors-23-04720-f004:**
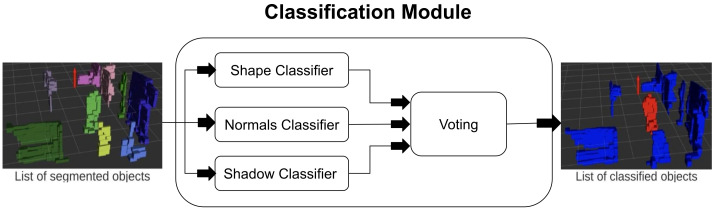
Flow diagram of the classification module. The module receives segmented objects as input and uses a voting scheme by three classifiers to classify them as either a person or not. Each object before classification is shown with different colors (on the left-most point cloud). The module also outputs a processed point cloud. The color red indicates a person and blue indicates otherwise.

**Figure 5 sensors-23-04720-f005:**
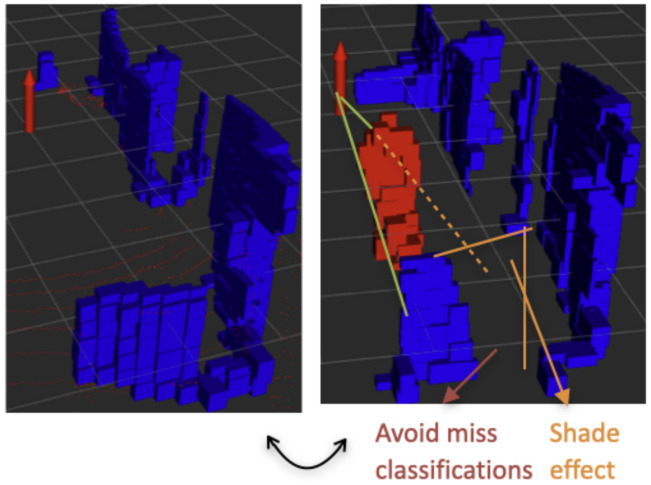
This Figure provides an example of how shading can impact object detection. The blue color represents the background while the red color represents a person. Shading can often lead to misclassification of objects.

**Figure 6 sensors-23-04720-f006:**
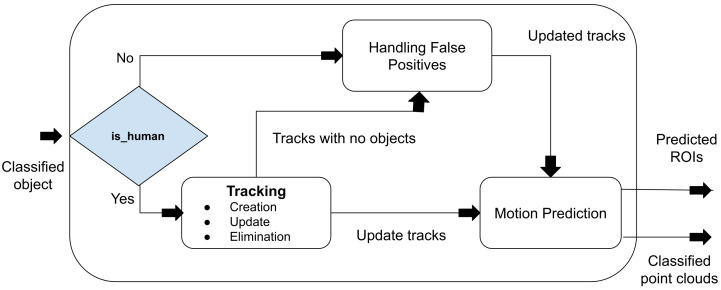
Flow diagram of the tracking module. Following person/not-person classification, the objects are fed into the tracker. The tracker then creates, updates, or eliminates the tracks based on certain assumptions, as explained in the text, and finally predicts the motion of the target person.

**Figure 7 sensors-23-04720-f007:**
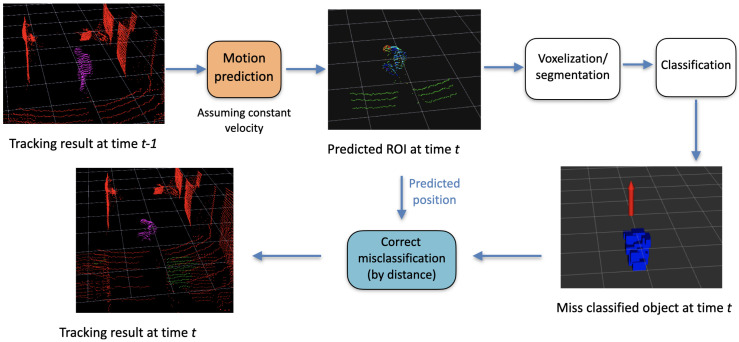
Abstract conceptual flow diagram for removing false positives and false negatives.

**Figure 8 sensors-23-04720-f008:**

Tracking results for extreme pose changes, ranging from crouching to doing jumping jacks, and holding an object. Red points were classified as background and violet points as a person.

**Table 1 sensors-23-04720-t001:** Sensor data acquisition with different cases.

Cases	Description
Case 1:	Single person is walking in front of the camera.
Case 2:	Single person is walking in front of the camera when there is occlusion.
Case 3:	Single person with different and complex poses.
Case 4:	Two persons are walking in front of the camera.
Case 5:	Three person walking in front of the camera with occlusion and doing complex poses.
Case 6:	Two persons are walking in front of the camera and the tracker tracks only the one person.

**Table 2 sensors-23-04720-t002:** Framework evaluation under different cases.

Case	Precision	Recall	F1 Score	Freq
Case 1:	95.37	95.12	95.25	9.01
Case 2:	90.18	72.24	80.22	7.84
Case 3:	94.88	94.49	94.68	8.61
Case 4:	93.65	95.16	94.40	6.80
Case 5 with normal walk	96.41	78.83	86.74	8.11
Case 5 with complex poses	97.51	91.35	94.33	7.71
Case 5 with occlusion	91.23	87.93	89.55	7.37

**Table 3 sensors-23-04720-t003:** Comparison of the proposed framework with the baseline method which is a publicly available package of ROS at GitHub. Adaptive clustering (AC) is also used for finding the ROI (segmented objects) and passed to our proposed framework.

Method	Precision	Recall	F-Score
Online Learning [31]	73.4	96.50	83.01
Proposed Framework	93.7	87.3	90.24
AC + Proposed Framework [13]	63.12	96.50	76.23

**Table 4 sensors-23-04720-t004:** Bandwidth comparison between baseline and the proposed method.

Method	Average Bandwidth (MB/s)	Mean Bandwidth (MB)	Min Bandwidth (MB)	Max Bandwidth (MB)
Online Learning [31]	16.59	1.69	1.67	1.72
Proposed Framework	9.06	0.90	0.90	0.91

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
