# Peer review of "Efficient Detection and Tracking of Human Using 3D LiDAR Sensor"

_sensors, 2023, doi:10.3390/s23104720_

Round 1

Reviewer 1 Report

The authors propose to approach, through LiDAR Technology, the problem of

 Detection and Tracking of Moving Objects in autonomous robots.

The authors present a framework for people detection, classification and tracking solution in an indoor/outdoor environment.

The proposed framework is divided into  motion detection, voxelization, segmentation, classification and tracking modules.

The authors state in section 3.3 that “assuming that non-moving objects are the background”, that is, what would happen if two people move? would this affect? Or how would it affect?

The authors state in section 3.4.1 of tracking that “we ensure that a track is only created when the same object is classified as a person by the classification module in three consecutive scans”, the question would be: How to know that in 3 consecutive scans is the same object? Since passing by another identified object could lose tracking if there's no way to ensure it's the same object, how do authors assume it's the same?

Although the characteristics of the experiments are described, the authors should make the data public so that the experiments can be replicated.

Although the authors show metrics like recall, F-measure, etc., there really isn't a comparison with other methods. This comparison should be in order to test the precision with respect to other methods and give greater validity to the results.

In the conclusions section, the authors affirm that "... integrated with a tracking mechanism that effectively handles extreme pose changes of individual", however, they state that characteristics of people should be assumed, such as their height of no more than 2 meters, that the width should not be greater than height, so I do not think it can be robust to position changes.

 In conclusion, I believe that the validity of the results depends on two things. The first is that they do not handle a public dataset so that it can be experimented with other methods. The second is that they are not compared to other techniques to test whether this proposed framework really works. outperforms other approaches

Regarding the wording, the authors should be consistent in citing a figure, for example in line 8 they refer to “Figure 1” and as “Fig 2” in line 95. Even in line 238 they cite as “Fig. 8” and on line 270 “Figure 8”

Author Response

We would like to thank the handling editor and referees for their careful reviews. We received major revisions based on those comments, and the comments have been very thorough and useful in improving the manuscript. We have taken them fully into account in the revision. We are submitting the updated manuscript with the suggestion incorporated into the manuscript. The response file is attached below. 

Reviewer 2 Report

This paper proposed a real-time approach for person recognition using 3D LiDAR data. The novelty sounds acceptable. However, I would recommend for publication after major revision.

(1)   Figures 3, 4, and 6 are stretched. I suggest to adjust this issue for better illustration.

(2)   Tables should be format consistent with template.

(3)   The related works, especially published in recent three years, should be discussed in introduction. As I can see, the current version is insufficient.

(4)   The comparative experiments are missing. 

Author Response

(The authors gave the same response as above.)

Reviewer 3 Report

This presents a framework for detecting and tracking persons using a 3D Lidar sensor.  The topic is fine, and the paper is well-structured with a good flow, ensuring coherence. The research findings are presented by lots of figures and tables. However, the logical presentation of the paper's achievements needs improvement, and some significant and minor revisions are required. Therefore, the following suggestions are made.

1)  The meaning of "Efficient" in the title of this study is not well reflected in the paper. It is necessary to indicate that the algorithm is efficient compared to others in terms of time and memory.

2) A discussion section is needed on how to determine if a person is set to be stationary for the background. Also the three classifiers in this paper are good enough to descriminate person?

3) It is not clear from the figures what they represent, especially regarding the location and position of people in Figures 4, 5, 6, 7, and 8. The text in Figures 4 and 6 is also difficult to read, and improvements are necessary.

4) Including photos for each case in Table 1 would help readers understand the situations better.

5) It would be helpful to use visualizations such as bar plots in Table 2 rather than simply presenting numbers.

6) The paper lacks a discussion section, and it would be beneficial to mention the limitations of the study.

7) The analysis of the results is weak, and it needs to be strengthened, particularly in addressing occlusion.

Author Response

(The authors gave the same response as above.)

Round 2

Reviewer 1 Report

Thanks for the answers and in some sections, for the clarifications

Author Response

Thank you very much for your effort.

Reviewer 2 Report

Thanks for revision.

All my concerns have been addressed. I recommend for acceptance.

Author Response

Thank you very much for your effort.

Reviewer 3 Report

All comments were revised and well updated. It seems to be published. 

Author Response

Thank you very much for your effort.